# Use of Thromboelastography and Rotational Thromboelastometry in Otolaryngology: A Narrative Review

**DOI:** 10.3390/jcm11041119

**Published:** 2022-02-20

**Authors:** Mathew K. Marsee, Faisal S. Shariff, Grant Wiarda, Patrick J. Watson, Ali H. Sualeh, Toby J. Brenner, Max L. McCoy, Hamid D. Al-Fadhl, Alexander J. Jones, Patrick K. Davis, David Zimmer, Craig Folsom

**Affiliations:** 1Department of Graduate Medical Education, Navy Medicine Readiness and Training Command, Portsmouth, VA 23708, USA; 2Department of Internal Medicine, Indiana University School of Medicine, Indianapolis, IN 46202, USA; fshariff@iu.edu (F.S.S.); pjwatson@iu.edu (P.J.W.); 3Department of Biological Sciences, Vanderbilt University, Nashville, TN 37235, USA; grwiarda@gmail.com; 4Department of Biochemistry, Indiana University Bloomington, Bloomington, IN 47405, USA; asualeh@iu.edu (A.H.S.); maxmcco@iu.edu (M.L.M.); halfadhl@iu.edu (H.D.A.-F.); 5Department of Biochemistry, Indiana Wesleyan University, Marion, IN 46953, USA; toby.brenner@myemail.indwes.edu; 6Department of Otolaryngology, Indiana University School of Medicine, Indianapolis, IN 46202, USA; jonesalj@indiana.edu (A.J.J.); davispk12@gmail.com (P.K.D.); 7Cleveland Clinic Foundation, Cleveland, OH 44195, USA; diz.zimmer9@gmail.com; 8Department of Otolaryngology, Navy Medicine Readiness and Training Command, Portsmouth, VA 23708, USA; folsomcraig@gmail.com

**Keywords:** thromboelastography, rotational thromboelastometry, otolaryngology, head and neck surgery, head and neck oncology, microvascular free flap reconstruction, obstructive sleep apnea, tonsillectomy, adenoidectomy, adenotonsillectomy, epistaxis

## Abstract

In the field of otolaryngology—head and neck surgery (ENT), coagulopathies present unique diagnostic and therapeutic challenges. In both hyper- and hypocoagulable patients, management of coagulopathies requires intricate attention to the nature of hemostatic competence. Common coagulation tests (CCTs) offer only a snapshot of hemostatic competence and do not provide a clear insight into the patient’s real-time hemostatic condition. Viscoelastic tests (VETs) offer a holistic and concurrent picture of the coagulation process. Although VETs have found prominent utilization in hepatic transplants, obstetrics, and emergent surgical settings, they have not been fully adopted in the realm of otolaryngology. The objective of this manuscript is to provide an overview of the literature evaluating the current utilization and possible future uses of VETs in the field of otolaryngology. The authors performed a comprehensive literature search of the utilization of VETs in otolaryngology and identified applicable studies that included descriptions of viscoelastic testing. Twenty-five studies were identified in this search, spanning topics from head and neck oncology, microvascular free flap reconstruction, obstructive sleep apnea, adenotonsillectomy, facial trauma, and epistaxis. The applicability of VETs has been demonstrated in head and neck oncology and microvascular free flap management, although their pervasiveness in practice is limited. Underutilization of VETs in the field of otolaryngology may be due to a lack of familiarity of the tests amongst practitioners. Instead, most otolaryngologists continue to rely on CCTs, including PT, PTT, INR, CBC, fibrinogen levels, and thrombin time. Learning to perform, interpret, and skillfully employ VETs in clinical and operative practice can greatly improve the management of coagulopathic patients who are at increased risk of bleeding or thrombosis.

## 1. Epidemiology, Definition, and Utilization of VETs in ENT

The intricate nature of hemostatic competence permeates through the field of otolaryngology—head and neck surgery. The field encompasses major surgeries that have a predilection for dynamic hemorrhagic complications involving facial trauma surgery, microvascular free flap reconstruction, and head and neck cancer resection. Otolaryngology encompasses a broad range of patients, meaning practitioners encounter patients at risk for bleeding due to an array of factors, such as therapeutic anticoagulation. Furthermore, some patients may be more susceptible to thrombosis due to an underlying malignancy or due to social and environmental risk factors. With a wide range of potential etiologies for coagulopathy, otolaryngologists must have intimate knowledge of resuscitation and hemostasis strategies [1]. However, perioperative hematologic evaluation is a topic largely unexplored in the otolaryngologic bodies of literature [2].

Viscoelastic tests (VETs) such as thromboelastography (TEG) and rotational thromboelastometry (ROTEM) have long been methods of guiding blood component therapy (BCT) and pro-hemostatic agents in hepatic and cardiac transplantation [3,4]. Recently, VETs have been widely adopted for the diagnosis and management of trauma-induced coagulopathy and bleeding conditions [5,6,7]. Although there is emerging literature on the utilization of VETs in otolaryngologic surgery, the purpose of this manuscript was to review the current and future applications of VETs in ENT.

### 1.1. Inadequacy of Common Coagulation Tests Specific to ENT

Prothrombin time (PT), international normalized ratio (INR), activated partial thromboplastin time (aPTT), Clauss fibrinogen concentration, and platelet count are examples of common coagulation tests (CCTs). However, these CCTs are limited in their ability to assess hemostatic competence, and most notably fail to assess platelet function, Factor XIII activity, and von Willebrand’s factor. This is a significant limitation, as von Willebrand’s disease and other platelet function disorders are common within the general population. Furthermore, elderly patients on anticoagulation regimens and patients who have large intraoperative hemorrhage or chronic inflammatory states may also pose increased risk for bleeding; yet, these conditions are not well evaluated with CCTs [8]. For these reasons, broad screening of INR/aPTT and platelet count are insufficient methods of identifying patients with an elevated bleeding risk while undergoing head and neck surgeries [9,10,11,12]. To guide transfusion of blood products, Thiele et al. recommends the assessment of dynamic coagulation factor consumption and coagulopathic risk [1]. CCTs are not well suited to evaluate this dynamic state, however, VETs are suitable to monitor the real-time dynamic factors affecting a patient’s hemostatic competence. The utility of VETs stems from their ability to provide a broader assessment of the blood clot lifespan, starting with initiation, amplification, propagation, and to eventual lysis/remodeling of the clot. The ability of VETs to provide the physician with an abundance of real time information about the patient’s hemostatic state is not offered by CCTs [5,13].

### 1.2. Principles of TEG and ROTEM

The earliest use of VETs was as a holistic assay to analyze hemostatic function. These tests are engineered to utilize rotation of either the cup (in TEG) or the pin (in ROTEM) to measure tension as an indicator of clot strength from clot initiation to clot lysis [5].

#### 1.2.1. TEG

When performing an assessment using TEG, a cup containing 0.36 mL of blood rotates 4.45 degrees at 10 *s* intervals. As coagulation begins, a bond between the fibrin and the pin is initiated and the tension on the wire increases. This tension is a function of clot strength, which is plotted on the *y*-axis and time is displayed on the *x*-axis. The plot provides a depiction of the clot’s evolution. A normal example tracing of a TEG graph is demonstrated in Figure 1. The TEG variables include reaction time (R), clot kinetics (K), α-angle, maximum amplitude (MA), and lysis at 30 min (LY30) [5,14]. ROTEM has a similar graph framework and the nuance between TEG and ROTEM will be discussed further in the next section.

#### 1.2.2. ROTEM

When comparing TEG and ROTEM, the key difference is that with ROTEM, the cup is stationary, and the pin rotates. The same measurements and tracing are created by both tests. However, the ROTEM utilizes a different set of nomenclature for each variable. For instance, clot time (CT) corresponds to R in TEG, clot formation time (CFT) equates to K, and maximum clot firmness (MCF) is analogous to MA. Clot amplitude at 5 and 10 min (A5/A10), measured at 5 and 10 min after the end of CT, respectively, is only measured in ROTEM. Lysis index at 30 min (LI30) is not the same as LY30. LI30 represents residual clot firmness 30 min following CT, depicted as a fractional percentage of MCF. Maximum lysis (ML) denotes the lysis detected as a percentage of the MCF during the run time and at the conclusion of the test [5,16,17].

#### 1.2.3. Analysis of ROTEM/TEG

The value of TEG and ROTEM is their ability to guide the transfusion of blood products tailored to a patient’s specific requirements and provide the physician with information that may necessitate an alteration in BCT. For example, if a patient’s tracing shows a long R/CT in hyperfibrinolysis, then fresh frozen plasma (FFP) is used. In the setting of a decreased α-angle, cryoprecipitate or fibrinogen concentrate is transfused. If the patient demonstrates a narrow MA/MCF, then platelets or fibrinogen are given [18]. When the LY30/ML is increased or the LI30 is decreased, then tranexamic acid (TXA) can be given [5,19,20]. Collectively, these parameters provide clinicians a comprehensive evaluation of coagulation which can be used to evaluate and treat a patient’s coagulopathy [21].

### 1.3. Adoption of VETs in Otolaryngology

Although viscoelastic testing has been adopted in various other surgical settings, including cardiovascular surgeries, liver transplants, and obstetrics [22], there is a dearth of literature regarding VETs in the realm of otolaryngology. VETs can be used to monitor patient coagulopathies while simultaneously allowing for point-of-care resuscitation assessment. Moreover, they help to prevent volume overload, which can often worsen coagulopathies when trying to correct hemorrhagic shock [5,23,24,25,26]. Additionally, hemostatic adjunctive therapy (HAT), with agents such as prothrombin complex concentrate, cryoprecipitate, soluble fibrinogen, TXA, and desmopressin, can be guided by VETs for anticoagulant reversal in emergent scenarios [27].

As previously discussed, VETs play a valuable role in BCT and HAT in severely hemorrhagic patients requiring massive blood transfusion. These patients are encountered during the practice of an otolaryngologist, whether in the operative theater, in the medical ward, or within the intensive care unit. Knowledge and implementation of VETs by the care team can guide, with greater accuracy, the replacement of blood components with FFP, prothrombin complex concentrate (PCC), cryoprecipitate, fibrinogen, and platelets. For example, R/CT and α-angle characteristics can guide the dosing of PCC, FFP, and cryoprecipitate. Additionally, analyzing VET values with particular emphasis on the R/CT can help to establish a baseline to follow when dosing PCC [20]. Furthermore, patients on antiplatelet drugs have higher rates of postoperative hemorrhage and are sometimes encountered by the otolaryngologist [28,29,30]. In these instances, MA/MCF parameters can guide platelet infusion [5]. For patients on anticoagulation medication who require emergent surgery, treatment with specific antidotes such as protamine for heparin, FFP and PCC for warfarin, and specific direct oral anticoagulant (DOACs) reversal agents can reverse coagulopathy. These can be administered with better precision from specialized VETs [31,32,33]. Thus, TEG and ROTEM provide expedient and effective point-of-care assessment of the need for anticoagulant reversal, demonstrating the utility of VETs in emergent settings [34].

There is significant opportunity in otolaryngology to adopt and utilize VETs in clinical and operative practice. Areas of use include head and neck oncology, microvascular free flap reconstruction, obstructive sleep apnea (OSA), adenotonsillectomy, facial trauma, and epistaxis. The otolaryngologist will encounter patients with both hypercoagulable states and hypocoagulable states, from both acquired and congenital coagulopathies. Furthermore, coagulopathic patients are commonplace in the subset of head and neck cancer patients [35,36,37,38]. Strict protocols and guidelines to guide blood component therapy using VETs do not yet exist in otolaryngology. However, protocols will not become commonplace until there is increased utilization and adoption of TEG and ROTEM in ENT. The remainder of this manuscript will be dedicated to reviewing the current body of literature and providing a summary of the adoption of VETs in otolaryngology.

## 2. VET in ENT: Current Literature

A comprehensive literature search of Ovid, MEDLINE, and the Cleveland Clinic Library databases was performed to assess the utilization of VETs in otolaryngology. This query consisted of keywords including ‘thromboelastography’, ‘rotational thromboelastometry’, ‘viscoelastic’, ‘otolaryngology’, ‘ear-nose-throat’, ‘head and neck surgery’, ‘head and neck cancer’, ‘microvascular free flaps and reconstruction’, ‘facial trauma and resuscitation’, ‘tonsillectomy’, ‘adenoidectomy’, ‘epistaxis’, ‘sinus surgery’, ‘oral and maxillofacial’, ‘free tissue transfer’, ‘free flap’, ‘microsurgery’, ‘endonasal’, ‘craniofacial’, ‘skull-based’, and ‘coagulation’. Inclusion criteria was determined by the authors if a study described the use of VETs in otolaryngology. A total of 670 studies were reviewed and of these, 25 studies met criteria for inclusion. These studies were further subcategorized into various areas within otolaryngology and are illustrated in Table 1.

### 2.1. Head and Neck Surgery

The otolaryngologist performs a wide range of procedures, ranging from routine outpatient clinic procedures to emergent tracheostomy in austere environments. It is a field that necessitates a keen understanding of human anatomy and pathophysiology. The head and neck has a robust blood supply, and as such, bleeding is not an infrequent encounter. The ability of VETs to quickly identify deficits in a patient’s primary and secondary hemostatic pathways and guide therapeutic hemostatic interventions in both the perioperative and postoperative period has the potential to supplement the otolaryngologist to better care for their patients. [35,62].

The potential benefits of VETs in otolaryngology are three-fold. First, they can act as a safeguard for patients that go undetected by preoperative bleeding assessments. These screening CCTs have a low sensitivity in identifying patients with undiscovered coagulopathies. Vries et al. recently assessed the screening efficacy of preoperative questionnaires and lab testing for hemostatic aberrancies, concluding that for both patients with and without a bleeding diathesis, the questionnaire was not a useful screening tool and CCTs demonstrated poor discriminating power [63].

The second benefit of VETs in otolaryngology is that they address the current deficits in CCTs’ abilities to accurately identify perioperative bleeding risks in patients. VETs may be particularly helpful in those patients with negative bleeding histories by standard questionnaires. A major systematic review performed by The British Committee for Standards in Haematology showed that there was a poor positive predictive value and low likelihood ratio for bleeding in those with abnormal CCT results [64]. One systematic review evaluated the predictive value of preoperative PT or INR for excessive perioperative bleeding following invasive procedures. This study concluded that PT/INR are not predictive for excessive perioperative bleeding following central venous cannulation, femoral arteriography, transjugular liver biopsy, core liver biopsy, percutaneous liver biopsy, or bronchoscopy. There was insufficient evidence to draw conclusions about paracentesis, thoracocentesis, lumbar puncture, or kidney biopsy. These results align with a prior study that looked at the predictive value of these tests for identifying bleeding risks during biopsy, bronchoscopy, and angiography [65]. The third benefit is that VETs are an expedient test, with results available in under 6 min. Therefore, VETs can guide tailored treatment plans for coagulopathic patients at the point-of-care and are rapid enough to be used in emergent situations requiring timely interventions.

Within the current body of literature, the incorporation of VETs in head and neck surgery has been promising. They appear to be particularly beneficial in identifying the dynamic changes in coagulopathic states seen during complex head and neck procedures. Patients undergoing head and neck surgery often manifest dynamic intraoperative physiologic changes, like acid–base and hemodynamic shifts, which can lead to derangements of hemostasis. For example, Mao et al. performed a study on artificial capnothorax while performing a thoracoscopic resection of an esophageal carcinoma [43]. The usage of a capnothorax induces an acidotic state and, therefore, researchers wanted to assess the effects on their subjects’ coagulation profiles. The results of this study found that there were significant derangements in the coagulation profiles of these patients, which were believed to be a function of acidosis from hypercapnia which impaired the functionality of thrombin. These impairments were reversed once the acid–base derangement was corrected, thereby demonstrating that the control of a patient’s pH status intraoperatively could create a more hemostatic surgical field. Due to the limitations of PT and aPTT assays in assessing coagulopathic status in dynamic situations such as respiratory acidosis, Mao and colleagues found that TEG was a more accurate technique for monitoring their patients. They also found that intraoperative use of TEG can inform the surgeon of the reduction in both the rate of clot formation and intensity of clot agglutination [43].

Head and neck anatomy is complex and surgical procedures are often performed in close vicinity to large caliber blood vessels. Surgical missteps can occur, resulting in large-volume hemorrhage both intraoperatively and post-operatively. If these situations arise, they may necessitate massive blood transfusion protocols, for which VETs can be employed to better guide resuscitation [42]. As an example, EXTEM (a form of ROTEM), was used by Durila et al. to guide blood product transfusion in tracheostomy patients with coagulopathies. In this study it was noted that EXTEM technology was useful in distinguishing hyper- from hypo-coagulopathic states in these patients whose PT/INR values are elevated at baseline due to inflammatory states. By providing information beyond the limitations of CCTs, VETs offer the benefit of more targeted blood product transfusion therapy and reduction of unnecessary transfusions in the operating room [40].

### 2.2. Head and Neck Oncology

The incidence of head and neck cancer has been steadily increasing over the past quarter century [66]. In tandem, the volume of oncologic surgery in the head and neck has also increased. Head and neck surgeons must be comfortable managing the degree of hemostatic risk involved with operations performed near the great vessels. VETs are not commonly utilized for patients undergoing head and neck oncologic surgery. Currently, most otolaryngologists follow the guidelines from perioperative risk assessment tools and groups. Examples of this include employing venous thromboembolic risk assessment using the Caprini scale or adherence to guidance from the Thromboembolic Risk Factors Consensus Group [67]. This group recognizes mechanical and pharmacologic prophylactic treatments, but of note, currently omits the utilization of VET monitoring to predict thromboembolic complication or monitor the hemostatic characteristics of patients.

Despite the generally low risk of developing thromboemboli in routine head and neck surgery, a recent study found that almost 13% of head and neck oncology patients developed venous thromboemboli after head and neck surgery in the absence of prophylactic hemostatic therapy [68]. It is well known that patients with head and neck cancer have an underlying hypercoagulable state and require prophylaxis prior to surgery. VETs have been shown to predict thromboemboli in patients with hematologic and non-hematologic malignancies [36].

TEG and ROTEM monitoring can be utilized to predict, diagnose, and treat the coagulopathic state of patients undergoing surgical intervention for head and neck cancer. Although the adoption of VET monitoring in clinical practice is not yet mainstream, its utility has been demonstrated in cases, including the treatment of cervical metastasis from squamous cell carcinoma, papillary thyroid carcinoma, and esophageal squamous cell carcinoma [38,43,45].

Selective neck dissection is a procedure to treat cervical lymph node metastases from squamous cell carcinoma. This procedure requires skeletonization of the internal jugular vein, and manipulation of the internal jugular can lead to a hypercoagulable state. Thus, utilization of VETs in selective neck dissection can assist in detecting a coagulopathic state intraoperatively and better inform that surgeon on appropriate management of the patient through the post-operative period [45].

VET monitoring has been shown to be a useful adjunct in the treatment of papillary thyroid carcinoma. Being often under-looked, patients with papillary thyroid carcinoma are often hypercoagulable due to atypical levels of coagulation factors and fibrinogen. Additionally, it was found that TEG parameters (coagulation index (CI), thrombogenic potential index (TPI), and angle) could potentially serve as diagnostic indicators for the detection of papillary thyroid cancer [38].

#### Microvascular Free Flaps

Microvascular free flap reconstruction has become a workhorse in head and neck oncology, with a high surgical success rate of 90–99% [46]. However, free flap surgery requires diligent surgical and postoperative care and any untoward complications following surgery can increase hospital stays and result in partial or total flap loss [51]. Currently, there are no well-defined protocols to identify patients that are at a high risk for postoperative flap loss. However, certain comorbidities, including COPD, tobacco use, and hypertension, have put certain patient populations at higher risk of developing abnormal coagulopathic states postoperatively [51]. The ability to prophylactically identify a patient with a high risk of coagulopathic complication would be exceedingly advantageous and would serve to improve patient care and surgical outcomes, as well as reduce healthcare resource burden.

TEG and ROTEM have shown efficacy in predicting and managing bleeding, yet have not been well studied in the field of microvascular reconstructive surgery. A study by Kolbenschlag et al. showed that pathological ROTEM parameters were strongly correlated with thrombotic flap loss [48]. Parker et al. showed that a functional fibrinogen-to-platelet ratio of ≥42%, as measured by TEG, can identify patients likely to develop thrombotic complications [46]. Vanags et al. demonstrated that in trauma patients who underwent surgery over > 30 days post-injury, TEG was able to detect hypercoagulability and predicted free flap thrombosis risk [52].

However, the use of TEG has not been entirely straightforward, with studies showing that its measurement does not correlate with coagulation [50,51]. The role of comorbidities and a lack of normalized standards for TEG and ROTEM make their applicability during use of VETs difficult. Consistently, functional fibrinogen-to-platelet ratio as measured by VETs has shown to be a good indicator of free flap thrombosis when compared to CCTs and other measurements [46,52]. However, all research in this field is in its infancy and further research will be required to create normalized values and applicability of test findings. Implementation of VETs stands to provide a considerable benefit to clinicians for preventing coagulopathic complications when undertaking microvascular free flap reconstruction.

### 2.3. Obstructive Sleep Apnea

Obstructive sleep apnea (OSA) is a condition wherein the patient has upper airway collapse, leading to lack of ventilation during sleep. OSA may be secondary to any disorder that compromises the upper airway including, but not limited to, obesity, retrognathia, micrognathia, adenotonsillar hypertrophy, or nasopharyngeal obstruction. Repetitive apneas lead to hypoxia, poor sleep quality, and excessive daytime fatigue. OSA is also associated with a wide array of metabolic and cardiovascular disorders that gradually increase morbidity and mortality. One of the main effects of OSA on the cardiovascular system is a deleterious effect on the patient’s hematologic profile, including a hypercoagulable state with elevated platelet aggregation and occasionally paradoxical embolism [55,56,69,70].

VETs are a tool well suited for assessing these hemostatic derangements; however, their utility has not been fully incorporated in the workup and treatment for OSA. Employing VETs in the treatment for OSA has shown benefit in guiding continuous positive airway pressure (CPAP) therapy. For instance, TEG can be used to track the progression from a hypercoagulable state to a normalized state following two to four weeks of CPAP therapy [54]. Additionally, TEG has been used to observe the effects of OSA on platelet reactivity (or lack thereof) to antiplatelet therapies [55]. Incorporation of VETs into the clinical toolkit of an otolaryngologist or sleep medicine physician may prove useful in both observing the efficacy of treatments and tracking their effects on the thrombotic characteristics of the patient.

### 2.4. Adenotonsillectomy

Tonsillectomy with or without adenoidectomy is one of the most commonly performed surgeries. Although it is considered a routine surgery, it is not without postoperative complications. Postoperative hemorrhage is the most common complication [71]. Preoperative CCTs such as PT/INR, aPTT, and platelet count do not adequately stratify which patients are at risk for bleeding [1]. Currently, the gold standard in determining who may be at risk for a coagulopathy is to do an extensive risk assessment, followed by ordering CCTs on patients that are identified as high risk [1]. However, healthy children undergoing a tonsillectomy or adenoidectomy showed no correlation between bleeding risk and prolonged PT and aPTT values when compared to those with normal values [12]. Thus, a laboratory test that could provide high sensitivity to the detection of coagulation complications would be considered extremely beneficial to improving postoperative outcomes.

The use of VET in tonsillectomy has the potential to improve patient treatment, but has not yet been studied extensively. Postoperative pain following tonsillectomy is usually controlled by opioids, but recently, nonsteroidal anti-inflammatory drugs (NSAIDs) have been gaining favor as an adjunctive analgesic. However, NSAIDs can theoretically increase bleeding risk due to decreased platelet aggregation [57]. TEG was used to study the effect of diclofenac on bleeding after tonsillectomy. When compared to the baseline, a statistically significant increase in MA and a decrease in R was observed after administration of diclofenac [57].

TEG can further be implemented to identify specific coagulation abnormalities and determine which blood products to administer. A case report by Raffan-Sanabria et al. examined a patient with post-tonsillectomy bleeding. They found TEG abnormalities, including a long k and low α-angle which dictated a treatment with fibrinogen and low MA which identified a need for platelets. TEG successfully monitored the coagulopathic state of the patient and guided blood component therapy [58].

### 2.5. Facial Trauma

Facial trauma is another area where VETs could play an important role. Facial trauma is a common reason for presentation to the emergency room and often requires operative repair [72]. These injuries can range anywhere from animal bites to complex craniomaxillofacial fractures caused by motor vehicle collisions or ballistic trauma. These injuries are often complex and involve a complex architectural framework consisting of dermal tissue, blood vessels, nerves, paranasal sinuses, and the upper aerodigestive tract, which make treating this trauma challenging [73]. Hemorrhage is not uncommon in these trauma patients, and their hematologic status is typically assessed with CCTs. These tests take time to obtain results and are not ideal for aiding in the correction of a coagulopathy in real time [74]. Due to the urgent time requirements when treating a trauma case, blood products are often given on an empirical basis following unsuccessful attempts at crystalloid resuscitation. Additional rounds of transfusion following acute stabilization would be more amenable to using the results of CCTs, hemoglobin, and hematocrit to guide the correct products to administer [75,76]. CCTs’ shortcomings are further compounded by the fact that, unlike VETs, CCTs are rendered inaccurate by blood transfusions. Because of this, CCTs have limited usefulness during trauma situations where massive blood transfusions are required [35]. VETs may provide a more convenient point-of-care test in the trauma setting that can improve upon empiric transfusion protocols or those guided by CCTs.

Several case reports have demonstrated the successful utilization of VETs in guiding BCT to acquire hemostasis following facial trauma. One study evaluated a cyclist who suffered multiple craniofacial fractures from a collision with a vehicle [60]. The patient became profoundly anemic during the treatment course. Even after the transfusion of red blood cells, colloid, and FFP, the patient was still unable to achieve hemostasis, and after the treating team ruled out both abdominal and thoracic bleeding, ROTEM was utilized to assess the coagulopathic state of the patient. EXTEM analysis showed that CT and CFT were prolonged and MCF was decreased. FIBTEM, a thromboelastometric test of fibrin-based clotting, showed both decreased A10 and decreased MCF. These findings indicated that the patient was suffering from a severe coagulopathy and the ROTEM findings guided administration of PCC, fibrinogen, platelet concentrate, and TXA. ROTEM was performed 2 h after transfusion and revealed normalization of coagulation status and clinical hemostasis was achieved after correction of the hematologic derangement [60].

VETs can also benefit the practitioner in caring for patients with less common pathologies. For example, a case report describes a situation where an individual suffered a snake bite, fell into a 1.5 m pit, and suffered facial fracture and epistaxis, unable to be controlled via packing. In the hospital, it was determined that the patient had consumptive coagulopathy, possibly secondary to the snake venom. CCTs showed elevated PT and aPTT with decreased fibrinogen levels. Furthermore, ROTEM results indicated no platelet plug formation. The patient was subsequently treated with antivenom, packed red blood cells, FFP, cryoprecipitate, and TXA with follow-up ROTEM testing showing improvement in clot time and clot formation time. Overall, three ROTEM tests were conducted at one-hour increments during the treatment course until the patient had normalization of coagulation. The authors concluded that VETs could be effectively used in treating patients with coagulopathy secondary to envenomation [61].

Utilization of VETs by an otolaryngologist in facial trauma can be a helpful adjunct, however the number of facial trauma cases that present with massive hemorrhage is less than <1%. However, while severe hemorrhage is rare, the risk of mortality is 37% when severe hemorrhage occurs in the setting of complex facial trauma [77,78]. Due to these high stakes, the use of VETs may be of great interest to future otolaryngologists and trauma surgeons treating patients with massive hemorrhage. Further research must be conducted on the use of TEG and ROTEM in the treatment of massive hemorrhage in trauma to develop standardized protocols to guide implementation and treatment strategies.

### 2.6. Epistaxis

Epistaxis is one of the most common otolaryngologic emergencies that present to either primary care or the emergency room. While most nosebleeds are mild in severity, epistaxis can be life threatening, requiring hospitalization or operative intervention. The current standard of care does not involve routine coagulation studies in patients with epistaxis [79]. It is recommended that the investigation for potential hemostatic disorders should only be performed when clinically indicated and often in consultation with a hematologist [79]. In that subset of patients, VETs may offer improved therapeutic guidance compared with CCTs.

Depending on the mechanism of coagulopathy, VETs can assist in guiding the replacement of red cells, platelets, coagulation factors, PCC, FFP, cryoprecipitate, soluble fibrinogen, TXA, amicar, and desmopressin [80]. Often, in epistaxis patients with underlying medical conditions requiring specific replenishment of pro-coagulates, VETs can guide in the most judicious method of reversing a coagulopathy.

The literature regarding the use of VETs in epistaxis is extremely limited [59,81]. Additional studies are needed to elucidate whether VETs offer superior therapeutic guidance when compared with CCTs for the small percentage of epistaxis patients in whom further investigation and/or intervention is clinically indicated to achieve adequate stasis. There is insufficient data available currently to make a definitive decision on whether the benefits of VETs outweigh the costs in the treatment of epistaxis due to underlying coagulopathy.

Considering how frequently patients suffer from epistaxis and the healthcare burden associated with emergency room visits and hospital admissions, there lies substantial potential benefit to the healthcare system by looking into further research of VETs to improve patient care and outcomes.

## 3. Conclusions

Currently, there is a limited body of literature studying the utilization of viscoelastic testing in the field of otolaryngology. The current utilization of VETs in the areas discussed above are limited in scope and generalizability due to the existence of a paucity of studies. However, with the increasing awareness of the utility of this technology and a constant push of our healthcare system to innovate and improve patient care and outcomes, more studies should and will be conducted to evaluate VETs’ effectiveness in otolaryngology. With adoption and implementation of this technology, the entire care team can utilize VETs, which have the capacity to provide rapid assessment and targeted therapeutic regimens, to treat coagulopathic patients.

## Figures and Tables

**Figure 1 jcm-11-01119-f001:**
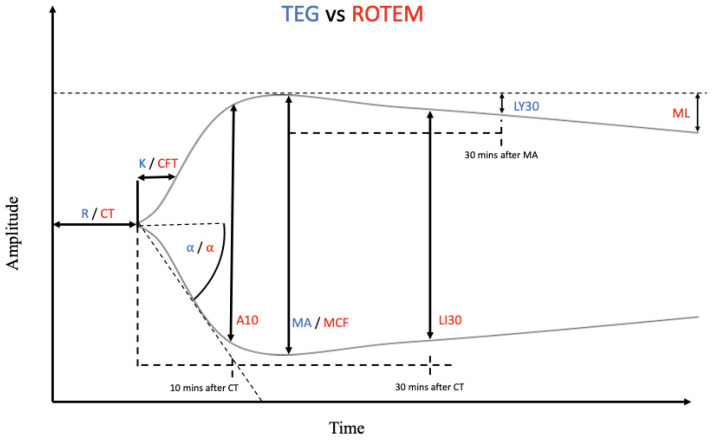
Depiction of a normal thromboelastography (TEG)/rotational thromboelastometry (ROTEM) tracing. The blue abbreviations represent the parameter labels for TEG, whereas the red abbreviations represent the parameter labels for ROTEM. Many of these parameters are representations of the same measurements between the two instruments, only with different nomenclature. Reaction time (R) and clot time (CT) are both the measurement of how long it takes for the transducer to be displaced 2 mm on the *y*-axis. Clot kinetics (K) and clot formation time (CFT) specify the amount of time it takes for the clot amplitude to grow to 20 mm after initial 2 mm clot growth. K and CFT represent initial clot strength and clot formation kinetics respectively. Alpha (α)-angle, in both TEG and ROTEM, represents the angle formed between the horizontal axis and the line formed from 0–20 mm amplitude. α-angle is calculated and utilized to measure the rate at which clot formation occurs. A10 is the amplitude reached 10 min after CT. Maximum amplitude (MA) and maximum clot firmness (MCF) refer to the clot’s maximum strength and are a measurement of the platelet–fibrin interaction. Lysis index at 30 min (LI30), a parameter unique to ROTEM, represents the percentage of MCF remaining 30 min after CT. Lysis at 30 min (LY30), a TEG parameter, refers to the percentage decrease in maximal clot amplitude 30 min after MA is achieved. Maximum lysis (ML) refers to the percentage decrease in clot amplitude measured at the end of the run [5,14,15].

**Table 1 jcm-11-01119-t001:** Summary of the literature of the use of VETs in ENT. ENT, ear nose and throat; INR, international normalized ratio; INTEM, intrinsic thromboelastometry; MA, maximum amplitude; OSA, obstructive sleep apnea; PT, prothrombin time; R, reaction time; TEG, thromboelastography; VET, viscoelastic test.

Discipline	Paper	Description
Head and Neck Surgery	Law 2001	Prospective randomized study: TEG was used to compare coagulation and blood loss associated with intravenous propofol infusion or inhaled isoflurane during anesthetic maintenance for head and neck surgery. Significant differences in either blood loss or coagulation were not observed between the two treatments [39].
	Durila 2015	Observational study: Tracheostomy was performed on 119 patients. INR showed that 55 patients had prolonged PT, but all TEG results were normal except one. Tracheostomy was safely performed on patients with normal TEG results without bleeding complications [40].
	Nguyen 2015	Retrospective study: Tested a transfusion algorithm/protocol for craniofacial reconstruction surgery by comparing pre-protocol and post-protocol cohorts. The protocol reduced intraoperative administration of blood products. The protocol utilized TEG to guide fresh frozen plasma transfusion due to the long turnaround time of INR [41].
	Klein 2016	Case study: Surgical intervention to stop massive arterial hemorrhage. Head and neck surgeon ligated common carotid using specific anesthetic strategy combined with ROTEM-guided massive transfusion protocol [42].
	Mao 2016	Prospective cohort study: Evaluated the impact of an artificial capnothorax on coagulation and fibrinolysis in patients undergoing thoracoscopic esophagectomy. Used TEG parameters to prove that these patients showed significant impairments in coagulation not observed in patients without artificial capnothorax [43].
	Mogensen 2017	Prospective study: Used TEG and common coagulation tests to evaluate a transfusion strategy in 40 infants [44].
Head and Neck Cancer	Mitchell 2005	Prospective pilot study: Studied 10 patients who underwent selective neck dissections to treat malignant disease of the head and neck. Measured coagulopathy using TEG. Found an insignificant increase in R [45]
	Nielsen 2013	Case report: TEG was used to determine that a patient undergoing removal of a malignant thyroid tumor was found to have abnormally increased hypercoagulable clot strength caused by tumor-induced upregulation of hemeoxygenase-1 [37].
	Lu 2020	Experimental non-randomized study: This study aimed to modify the TEG parameters for papillary thyroid carcinoma and nodular goiters. Using 62 nodular goiter patients, 53 papillary thyroid carcinoma patients, and 61 healthy patients, correlation analysis demonstrated hypercoagulable TEG parameters for papillary thyroid carcinoma patients [38].
Free Flaps	Parker 2012	Preliminary study: Used TEG to measure fibrinogen:platelet ratio to probe for post-surgical thrombotic complications. A functional fibrinogen to platelet ratio above 42% as measured by TEG may be useful in identifying those patients likely to develop thrombotic complication [46].
	Murphy 2013	Case report: Patient diagnosed with rhino-orbital mucormycosis underwent orbital exenteration, extensive cheek and sinus debridement, and reconstruction using a free myocutaneous anterolateral thigh flap. Preoperative TEG measured functional fibrinogen:platelet ratio. Intraoperative platelet administration was guided by TEG [47].
	Kolbenschlag 2014	Retrospective review: Reviews diagnostic value of ROTEM for screening patient vulnerability to thrombotic complications with specific regards to reconstructive microsurgery [48].
	Fuller 2015	Review: Description of risk factors in microvascular tissue transfer following ablative surgeries for head and neck malignancy. Patients with extrinsic coagulation pathway thrombus formation or intrinsic coagulation thrombus formation mean clot formation >72 mm or functional fibrinogen:platelet ratio >43 are at a significantly higher risk of thrombotic flap loss [49]
	Wikner 2015	Prospective, exploratory cohort study: Both standard testing and ROTEM/INTEM were incapable of predicting perioperative bleeding, thrombosis, or flap loss. Intraoperative use may prevent blood loss provided standardization can be achieved [50].
	Ekin 2019	Retrospective study: 77 patients had flap surgery and no relationship was found between preoperative or postoperative TEG results and flap complications. There was no correlation with flap loss and pre- or postoperative TEG results [51].
	Vanags 2020	Prospective observational study: 103 adults trauma patients who received flap surgery either <30 days post-trauma (ES group) or >30 days post-trauma (LS group). Postoperative hypercoagulability on ROTEM predicted free flap loss in the LS group but not the ES group [52].
OSA	Othman 2010	Animal model study: TEG performed on rats concluded that intermittent inspiratory occlusions cause transient hypercoagulability [53].
	Toukh 2012	Prospective crossover study: Used TEG to test the hypothesis that patients with severe OSA are hypercoagulable and that two weeks of continuous positive airway pressure reduces this hypercoagulability. Concluded that TEG can detect hypercoagulability in patients with OSA [54].
	Gong 2018	Cross sectional observational study: Tested whether OSA has effects on platelet function profiles in acute coronary syndrome patients on dual antiplatelet therapy. TEG platelet mapping assay was used to detect effects of antiplatelet therapy with arachidonic acid and adenosine diphosphate activators, which demonstrated that OSA-induced platelet hyperactivity persists despite antiplatelet therapy [55].
	Fernandez-Bello 2020	Observational study: ROTEM showed hypercoagulable state in patients with OSA due to increased platelet/leukocyte aggregation and endothelial damage [56].
Tonsils and Adenoids	Heaney 2007	Prospective observational study: Utilized TEG to monitor clot strength following the preoperative administration of diclofenac for pediatric patients undergoing tonsillectomy. Found to have a significant reduction in R and a significant increase in MA [57].
	Raffan-Sanabria 2009	Case report: Female patient suffered severe oral bleeding three weeks post-tonsillectomy. Adequate hemostasis and control of bleed were achieved through use of TEG-guided decision making [58].
Facial Trauma and Hemorrhage	Alesci 2011	Retrospective study: Analyzed 75 patients visiting a hemophilia center for bleeding tendency including dental bleeding, gum bleeding, and epistaxis. No ROTEM parameters were significantly out of normal range [59].
	Grassetto 2012	Case report: Cyclist suffered serious craniofacial trauma and massive hemorrhage status post motor vehicle collision. ROTEM-guided administration of prothrombin complex concentrate and fibrinogen concentrate was effective in correcting coagulopathy [60].
	Abraham 2020	Case report: Patient was bitten by hematotoxic snake, lost consciousness, and suffered facial trauma after fall. Discovered to have ongoing oromaxillofacial bleeding upon arrival at the emergency department. Underwent ROTEM-guided transfusion to correct coagulopathy along with mandibular repair [61].

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
