# Peer review of "Use of Thromboelastography and Rotational Thromboelastometry in Otolaryngology: A Narrative Review"

_jcm, 2022, doi:10.3390/jcm11041119_

Round 1

Reviewer 1 Report

The article reviews the use of thromboelastography and rotational thromboelastometry in the field of otorhinolaryngology.  

The article is well written and clear to understand.  Unfortunately, and this is not the fault of the authors, there are currently no randomized studies that support the routine use of these techniques for the treatment and optimization of the management of these patients. However, this narrative review summarizes the most important potential applications where this technique could be useful in the area of otorhinolaryngology.

Two misspellings

  1. Acronyms should always be defined the first time they are cited in the article

Page 4, Line 160

Direct oral anticoagulants (DOAC)

  1. Page 8, Line 266

It should be VETs instead of VTEs

Reviewer 2 Report

The authors of the manuscript focused on rotational thromboelastometry a  detect anticoagulant drugs activity.  The authors have come up with an interesting review that is very important for medical research. Rotational thromboelastometry and thromboelastography is a holistic blood coagulation assay.  Rotational thromboelastometry is a viscoelastic hemostatic assay that has been used in emergencies (trauma and obstetrics), and surgical procedures (cardiac surgery and liver transplants), but experience with this assay in anticoagulant-treated patients is still limited. The positive thing about the manuscript is that the authors focus on its use of viscoelastic hemostatic assays in otorhinolaryngology The manuscript is well structured, but some parts of the manuscript need to be corrected and supplemented in order for this manuscript to be published.

Page 2, lines 57-62: The authors present the clinical practical use of ROTEM in various clinical situations. In this section, the authors should state that it is also used in the diagnosis of bleeding conditions. These data were published in a recent review. This manuscript should be quoted by the authors.: Diagnostics 2021, 11(11), 2140; https://doi.org/10.3390/diagnostics11112140.

Pages 3-4, lines 125-134: MA/MCF are important values that are critical in the use of fibrinogen concentrate in patients with fibrinogen disorder. This knowledge was also used in demanding perioperative management. We know about special clinical states for severe coagulopathies, where in perioperative management the patient must be given anticoagulation and coagulation treatment due to the high risk of thrombosis. Perioperative management of hypofibrinogenemic patients is complicated, requiring a specific assessment of the patient's overall hemostasis, taking into account both the bleeding and thrombotic risk. ROTEM can assist in the management, this was published in a manuscript that the authors should cite: Thromb Res. 2020 Apr;188:1-4. doi: 10.1016/j.thromres.2020.01.024. 

Table a figure in the text are very clearly written.

I have to say that with these 77 references of which less than half of the references are newer than 5 years. It would be appropriate to use more new references.

Reviewer 3 Report

Viscoelastic tests (VETs) such as thromboelastography (TEG) and rotational thromboelastometry (ROTEM) have long been a method of guiding blood component therapy (BCT) and pro-hemostatic agents in hepatic and cardiac transplantation with recent wide-spread adoption in trauma resuscitation. As the authors wrote, there is emerging literature on the utilization of VETs in otolaryngologic surgery. Therefore, the purpose of this manuscript is to review the current and future applications of VETs in ečar, nose and throat (ENT) region.

The manuscript fullfills the criteria of the review article. It contains the review of the current data on the use of VETs in ENT surgery, oncology, in the case of obstructive sleep apnea, adenotonsillectomy, facial trauma and epistaxis that may be associated with disorder of heaemostasis. 

Thus, I consider the manuscript as the useful contribution to the clinical management of the patients undergoing procedures in the ENT region.   

Round 2

Reviewer 2 Report

The presented manuscript has been corrected in response to the suggestions. The authors have followed the recommendations of the reviewer. After the revision, the provided data and addition of the results became more clear. I would like to thank the authors for resubmitting the manuscript and explaining the obscure points from the previous version.